# Attributes of Organizational Health Literacy in Health Care Centers in Iran: A Qualitative Content Analysis Study

**DOI:** 10.3390/ijerph19042310

**Published:** 2022-02-17

**Authors:** Elham Charoghchian Khorasani, Seyedeh Belin Tavakoly Sany, Mehrsadat Mahdizadeh, Hassan Doosti, Hadi Tehrani, Mohammad Moghzi, Alireza Jafari, Nooshin Peyman

**Affiliations:** 1Department of Health Education and Health Promotion, Faculty of Health, Mashhad University of Medical Sciences, Mashhad 9137673119, Iran; echaroghchian@yahoo.com (E.C.K.); tavakkolisanib@mums.ac.ir (S.B.T.S.); mahdizadehtm@mums.ac.ir (M.M.); tehranih@mums.ac.ir (H.T.); 2Department of Mathematics and Statistics, Macquarie University, Sydney, NSW 2109, Australia; hassan.doosti@mq.edu.au; 3Public Health Expert, Health Center No. 2, Mashhad University of Medical Sciences, Mashhad 9137673119, Iran; moghzim1@mums.ac.ir; 4Department of Health Education and Health Promotion, School of Health, Social Development and Health Promotion Research Center, Gonabad University of Medical Sciences, Gonabad 9P67+R29, Iran; jafari.ar94@gmail.com

**Keywords:** health literacy, organizational health literacy, health care centers, health promotion, public health, health education

## Abstract

Organizational Health Literacy (OHL) is described as a new concept to remote health organizations to implement practices, policies, and systems that make it easier for patients to use, understand, and navigate health information to take care their own health. In Iran, there is no consensus on the attributes of OHL, and its practical implications and scope have not been evaluated. This manuscript is one of the first attempts to explain the attributes of the OHL in health care centers in Iran. This study is a content analysis survey, which was guided by the attributes of the OHL provided by Brach et al. and 26 semi-structured interviews were conducted with Iranian health professionals and employees of healthcare organizations from June 2020 to January 2021. A data analysis was performed using the MAXQDA 10 software. Across the study, ten sub-themes, 21 subcategories, and 67 codes emerged. The 10 main attributes of OHL were management, integration of health literacy in the organization, workforce, participation, range of HL skills, HL strategies, access, media variety, the role of the organization in crisis, and costs. These attributes may guide the planning of health care centers improvements and have the potential to promote health service reforms and public health policy.

## 1. Introduction

In 1974, the concept of health literacy (HL) was introduced [1] and it was defined as “the cognitive and social skills, which determine the motivation and ability of individuals to gain access to, understand and use information in ways, which promote and maintain good health”. Defined this way, HL goes beyond the concept of health behaviors and health education-linked communication, and addresses the political, social, and environmental characteristics that influence community health [2,3].

In Iran, the first study on HL was related to 2007. In this study, the TOFHLA questionnaire was used to investigate the level of HL in the Iranian population. The results of the study showed that 71.9% of the population participating in this study had inadequate and borderline levels of HL [4]. Therefore, the attention of health researchers was drawn to HL, and many studies on HL were subsequently conducted in Iran [5,6,7]. Researchers designed and localized the HL questionnaires, and adopted intervention measures to promote the HL of different groups of people [8]. In addition to the above, scientific conferences and congresses focusing on HL were held in Iran, and different journals on the HL concept were established [9,10]. However, according to a newly published review study in 2021, insufficient HL is common in the Iranian population [7,11]. Their finding highlighted some of the major challenges and gaps in the field of HL in Iran such as the lack of a clear and fixed definition for HL, the lack of OHL-related attributes and guidelines in health services to manage patients based on their HL levels, no locally comprehensive screening instrument for HI in Iran, and poor patient-health provider communication skills [7,12]. Therefore, promoting HL in Iran is necessary to provide the base on which the Iranian population are enabled to improve their own health through making better decisions related to their health condition, involve successfully with self-care skills and health community action, and push authorities to meet their duties in addressing health equity [5,7]. Meeting the HL needs in Iran will promote progress in decreasing health inequity. Efforts to improve HL are crucial in whether the environmental, economic, and social promotion are fully realized [12].

Several studies have shown HL activities should be considered in the health care organization structure [13,14,15]. Organizational Health Literacy (OHL) are organizations that “facilitate the guidance, understanding, and use of information and services for health care” [16]. Although, the OHL concept started in 2012 and several researches were conducted on OHL’s concept, there are not enough studies on the implementation of OHL structure [13]. Brach et al. expressed ten attributes for OHL [3], which are used by several tools related to OHL. These attributes include: (1) leadership that makes HL integral to its mission, structure, and operations; (2) integrates HL into planning, evaluation measures, patient safety, and quality improvement; (3) prepares the workforce to be health literate and monitors progress; (4) includes populations served in the design, implementation, and evaluation of health information and services; (5) meets the needs of populations with a range of HL skills while avoiding stigmatization; (6) uses HL strategies in interpersonal communications and confirms understanding at all points of contact; (7) provides easy access to health information and services and navigation assistance; (8) designs and distributes print, audiovisual, and social media content that is easy to understand and act on; (9) addresses HL in high-risk situations, including care transitions and communications about medicines; and (10) communicates clearly what health plans cover and what individuals will have to pay for services [3].

It is evident that HL is not still integrated into the health organization’s strategic planning and vision in Iran. Therefore, health policymakers and researchers face more challenges in what they recognize about attributes of the OHL and what steps they should use in OHL-related attributes to improve HL skills [6,7,17]. Since no action has been taken in Iran for the OHL, this study aims to explain the attributes and content of the OHL in health care centers in Iran. Such information helps health services to effectively implement OHL-related attributes to improve the quality of health service and personal health literacy skills in the community.

## 2. Materials and Methods

### 2.1. Study Design

In this study, we implement an exploratory design and qualitative approach of semi-structured interviews to understand the health professional’s perspectives about the attributes of the OHL in Iran. The theoretical framework was designed based on ten OHL-related attributes, which were expressed by Brach et al.

### 2.2. Study Setting

This study was conducted on 26 participants selected from different cities (Mashhad, Tehran, Tabriz, Yasuj, Maragheh, Bojnourd, Zahedan, Torbat Heydariyeh, Kurdistan, and Chenaran) from June 2020 to January 2021. Owing to the outbreak of COVID-19, the interview was conducted over the phone. Twenty-six interviews were conducted with health professionals and health center staff from different cities in Iran.

### 2.3. Participants

The study participants were selected from different medical and healthcare specializations such as health educators, health service managers, healthcare providers, physicians, pharmacists, nurses, and midwives. A security guard who was also working in healthcare centers was selected to participate in this study. We used the target-based sampling method and the snowball method with maximum variability to find eligible participants. Sampling was gradually continued until the data were saturated (Table 1).

### 2.4. Data Collection Tools and Procedures

The interview records were designed based on the attributes of OHL that were defined by Brach et al.’s [3]. All interviews were conducted by telephone because of the COVID-19 pandemic. The researcher made telephone calls to the participants and explained the objectives of the study to them. A specific time was then set for the interviews. An informed consent form and interview guide questions were emailed to participants. Interviews were recorded with the permission of the participants, through a special software installed on the researcher’s mobile phone. The average interview time was 45 min.

### 2.5. Research Questions

The researcher’s first question from the participants was, what do you think about the OHL-related attributes? Then, semi-structured questions were used during the interview. The semi-structured questions were designed according to the attributes of Brach et al. as follows:How do we involve the management of the organization in HL?How do employees prepare to carry out HL activities in the organization?How can people participate in the organization?What features do you think the media should have that are easy for people to read, understand and practice?How can we be sure that people have understood our content?How is HL the focus of an organization’s activities?How to use HL skills to meet people’s needs?What factors or conditions do you think facilitate access to health information and services?How can the organization improve the HL of people in dangerous medical and nursing conditions?How can HL be useful for information on health insurance and health care coverage and costs?

### 2.6. Data Processing and Analysis

The Lundman and Graneheim method was used for analysis [18]. The text of the interviews was implemented verbatim. The text was summarized into semantic units and the initial codes were extracted. Then, the abstract codes were formed according to the participants’ experiences, which led to forming overt and covert concepts. All codes were categorized into sub-categories; therefore, the main themes were formed. We used MAXQDA version 10 to conduct data analysis, coding, and the extraction of categories and themes. In order to achieve trustworthiness (transferability, credibility, and reliability), measures were taken in the steps of the research process.

## 3. Results

The study involved 26 people, half of whom were female interviewees and other participants were members of health faculty (Table 1). In this study, 57.6% of participants had a PhD degree and 88.6% of participants were specialists in the field of health education. The average work experience of the participants was 16.8 ± 1.8 years, ranging from 4 to 33 years. Participants from 10 cities (Mashhad, Tehran, Tabriz, Yasuj, Maragheh, Bojnourd, Zahedan, Torbat Heydariyeh, Kurdistan, and Chenaran) attended this study. The characteristics of the interviewees are shown in Table 1.

The average duration of the interview was 44.8 ± 13.04 min, ranging from 25 min to 80 min. Figure 1 shows the attributes of the OHL presented by Brach et al. and its changes in the current qualitative research. In 26 interviews, 895 initial codes were extracted from interviews. Analogous codes were then merged, and finally, during this inductive process, 67 codes were revealed in 21 subcategories, 10 sub-theme, and one theme. The main theme, sub-themes, subcategories, codes, and meaning units are shown in Table 2 and Figure 1.

## 4. Discussion

The present qualitative study was conducted to explain the characteristics of the OHL in health care centers in Iran. This qualitative study is based on the content analysis of the ten attributes of Brach et al. [3]. We identified 10 main attributes of OHL in Iran through depth 26 interviews with health professionals and health care center staff. These attributes include management, integration of HL in the organization, workforce, participation, range of HL skills, HL strategies, access, media variety, the role of the organization in crisis, and costs.

### 4.1. Manengment

Management is one of the main attributes of the OHL, which has been introduced as a leader in various studies [6]. In the present study, participants preferred to use the word “management” instead of using the word “leadership”, so to localize the attributes of the OHL, the word management was used instead of leadership. Several studies introduced leadership as the first attribute of OHLs in healthcare centers [6,17]. It has been also introduced as one of the attributes of the OHL in the tools or guidelines (AHRQ, OHLO, HLHO-10, VHLO, Org-HLR, and C-CAT) [6]. In this study, four subcategories were created for the leadership through the interviews including ‘appreciation of the staff implementing HL’, ‘encouraging other organizations to use HL’, ‘appointing a supervisor to implement HL’, and ‘handling people’s complaints’. Appreciating employees in the health organization plays critical role in the improving OHL structure. Borkowski believes that leaders can help employees to cope with change and help them strive to achieve their goals and incentives. Leaders in healthcare centers should be so that employees can easily share their point of view toward the OHL [19]. Another subtheme of managers in this study is to encourage other organizations to use HL. This encouragement is possible by providing reasonable evidence. Providing credible evidence is important to other organizations for two reasons: first, lack of evidence is considered as a major obstacle to the organization’s progress toward HL. Palumbo et al. indicated that inadequate evidence is the main obstacle to the organization’s transition to an OHL [20]. Second, the lack of compelling evidence has prevented health providers from considering the OH concept as a health priority [6,17]. For example, Cooper et al. pointed out that the lack of credible and reliable action reports prevented health policymakers and managers from accepting OHLs as a health priority [21]. Therefore, one of the important missions of OHL managers is to provide reliable reports to promote the transformation of the HEALTH organization into an OHL structure [6,17].

The leader has several responsibilities to develop the OHL structures, but the two tasks that emerged from the perspective of the participants in the present study were the appointment of a supervisor to oversee the implementation of HL in the organization and the handling of public complaints. The leader of the organization should appoint a person to supervise the HL activities in the organization so that the supervisor can follow the HL activities in the organization in a more specialized way. In the study of Brach et al., appointing a person in charge and delegating authority to oversee HL is one of the main tasks of the leader in an OHL [3].

### 4.2. Integration of HL in the Organization

The second attribute of the OHL is the integration of HL into the organization’s activities. In order to integrate HL into the organization, our research participants believe that changing both the structure and function of the health organization is essential to facilitate the integration of HL into the organization’s activities. Hence, two subcategories were created under the headings of ‘integrating HL in the structure of the organization’ and ‘integrating HL in the functioning of the organization’, and three codes were obtained, as follow: explaining HL in goals and statements clients health, the providing documents for the implementation of HL, and the evaluation of HL promotion activities.

Abrams et al. (2014) indicated that for the integration of HL into the organization’s structure, HL measures must be defined in the goals and statements of the organization [22]. Likewise, appropriate guidelines must be designed for improving the level of HL skills among employees and clients in the health organization because promoting HL in health workers not only improves the patient perception toward health information but also greatly increases patient-provider interactions [23,24]. Positive interactions between service providers and patients are known to as the main facilitator to improve patient health outcomes because patients who are satisfied with their healthcare providers are more likely to adhere to treatment plans and participate in improving their own health [25]. Likewise, documentation is the other main subtheme that shows HL activities are being implemented and evaluated in the health organization [6]. The results of a review study conducted in 2021 by Palumbo et al., show that although many barriers, such as lack of time and limited available resources, prevent the transition to an OHL, one of the steps that must be taken to overcome the existing barriers is to make more efforts to evaluate the OHL and to clarify its role in preventative medicine [26].

### 4.3. Workforce

Workforces of OHL’s play a key role in solving HL issues and helping to bridge the gap between limited HL and health-related outcomes [27,28]. The workforce in tools (AHRQ, OHLO, HLHO-10, VHLO, Org-HLR, and C-CAT) has also been introduced as one of the attributes of the OHL [6]. In this study, the workforce was identified based on two sub-categories of ‘training’ and ‘employment’. For the training sub-category, four codes were obtained under the headings of ‘HL training for employees’, ‘appointing the person in charge of HL training for employees’, ‘teaching communication strategies to employees’, and ‘employees’ access to HL training materials’. For the employment sub-category, three codes were obtained under the three headings of ‘hiring employees proficient in common language’, ‘hiring employees familiar with HL’, and ‘having HL in the employee job description’. “The OHL supports HL training for all health care workers”, said Brach et al. [3]. In most studies, the focus is more on training medical and nursing staff who are in close contact with clients [29,30]. Other health care workers have also been ignored. Therefore, although HL skills are important for nurses and physicians, all employees should receive HL training [23,24]. Our finding is consistent with several studies on HL that reported health workers usually lack adequate knowledge and skills to effectively communicate with patients with low HL [30,31]. Therefore, it is essential that health workers who work in health care systems have the adequate skills to effectively communicate with patients [19]. In addition, hiring people who are familiar with HL skills should be a priority. Brach et al. also believed that the organization should employ a different workforce with HL expertise [3]. As stated in the study by Brach et al., bilingual staff with HL skills should be employed in the OHL to assist clients whose language is different, and if this is not possible, an interpreter should be employed in the organization [3].

### 4.4. Participation

Another attribute of the OHL in Iranian healthcare centers is participation [6]. For the participation sub-theme in this study, two subcategories were obtained with the titles of public participation and employee participation, which covered five codes including “survey of clients regarding the choice of treatment methods’, ‘planning to employ volunteers and health liaisons’, ‘existence of instructions for employee participation in HL planning’, ‘existence of instructions for employee participation in implementing HL’, and ‘existence of instructions for employee participation in HL assessments’. Brega et al., highlighted patient participation as one of the conceptual frameworks of OHL [24]. It has been reported that patients with more participation are likely to make better health decisions and take appropriate steps to properly manage chronic health conditions [32]. There is evidence that increasing patient participation is important for improving health outcomes, especially for patients with limited HL skills [33]. In Iran, surveying clients in the choice of treatment methods is not a routine issue and most physicians make their own decisions and the patient is required to comply with it, which should be corrected in the OHL structure to increase patient participation. Various studies have shown that patients and organizations benefit from the active participation of patients to make better health decisions about their health [34]. Further, a guideline must be designed to increase employee participation in the planning, implementation, and evaluation of HL in the health organization. Borkowski (2011) pointed out that employee participation can also reduce resistance to change and help employees develop a positive view toward OHL. In addition, employee involvement may help sustain the effective use of the OHL’s actions [19].

### 4.5. The Range of HL Skills

The range of HL skills is another main attribute of the OHL in the present study, which was considered in several OHL tools (AHRQ, OHLO, HLHO-10, an Org-HLR) [6]. The sub-theme of HL skills in this study was identified with two subcategories under the headings of ‘essential HL skills in relation to clients’ and ‘needs assessment using HL skills’. These sub-themes included five codes: ‘welcoming and evaluating the needs of clients upon entering the organization’, ‘ensuring that the client’s needs met when leaving the organization’, ‘gather essential information from clients (only once in the organization)’, ‘identify and guide clients in need of additional assistance’, and ‘determining the HL status of clients’. Therefore, in the OHL, a series of essential HL skills should be considered, for example, the customer should be welcomed by the health staff when entering the organization and the client’s needs are measured upon entering the organization. For example, ask the patient, why were you referred to the organization? Or what is the problem? Then when leaving the organization, check if their needs have been met [3,6]. In the convenience for the client, their essential information (e.g., demographic information, and information related to medical and laboratory records) should be taken and registered in the system. Likewise, avoiding asking repetitive and boring questions is essential [30,31].

### 4.6. HL Strategies

The sixth attribute of the OHL in our study is the strategies of the OHL. The category of ‘HL strategies’ was identified based on two sub-categories of ‘HL strategies by employees in the field of verbal interaction with clients’ and ‘observance of HL strategies by employees in the field of support and writing with clients’. Likewise, eight codes were obtained under the headings of ‘using simple, clear, and understandable language’, ‘talk to clients with appropriate voice and medium speed’, ‘ensure you understand the content provided by clients by getting feedback’, ‘answering clients’ questions and encouraging them to ask questions’, ‘allocate enough time for each interaction’, ‘provide training clearly by stating the main and important points’, ‘provide training according to the economic ability of clients’, and ‘provide written training to clients according to their characteristics’ formed.

Adhering to verbal HL strategies is very important [27]. In order to improve the communication between patients and health providers, the US centers for disease control and prevention made the following recommendations in 2011: (1) use a simple language, use less medical terminology, (2) use the medical terms if there is no easier and more familiar term to replace, and (3) use feedback to ensure that clients understand [24,29]. Another code in this subcategory is answering clients’ questions and encouraging them to ask questions. Patients with limited HL are often embarrassed to admit that they do not understand and ask fewer questions [35]. Therefore, the staff of OHLs should have the characteristics of encouraging clients to ask questions. Encouraging clients to ask questions is one of the tools for promoting clients’ self-management and empowerment [36]. Brach et al. believe that HL campaigns should be launched to encourage questions, and that all questions should be answered satisfactorily [3]. Another code obtained in this study is to allocate enough time for each interaction. It was evidenced that patients are often misunderstood to interpret medical information when receiving a large amount of medical information within a few minutes after visiting a doctor [37,38]. In a study that examined how much information patients remember when visiting a doctor, found that the ability to maintain verbal health information is challenging, even for those with adequate HL skills [39]. Therefore, it is important to improve written communication because it is well-known that health materials help increase the understanding of health information and are useful for patients at all levels of HL, not just those with a low level of HL [40,41].

### 4.7. Access

Access is the seventh attribute of the OHL, which was obtained from this qualitative study. It is also used in five tools (AHRQ, OHLO, HLHO-10, VHLO, and Org-HLR) as one of the main attributes of the OHL [6]. Two sub-categories were obtained under the headings of ‘obtaining services and buildings’ and ‘obtaining understandable and executable information’; and seven codes were defined for the sub-category of “access to services and buildings” which includes: ‘easy access to information about the organization and services provided’, ‘use understandable boards and guides in the organization’, ‘appropriate number and arrangement of chairs in the organization’, ‘easy access to appointment scheduling methods and their accuracy’, ‘considering a suitable space for parking vehicles’, ‘possibility of clients’ access to valid educational resources’, and ‘sending the needy to help and charity centers’. Likewise, three codes were obtained for the category of “access to understandable and executable information” which are ‘design all forms and documents in simple and understandable language’, ‘evaluate all content distributed in the organization by getting feedback from clients’, and ‘preparation of forms and documents in the common language of the clients’.

Several studies reported that OHL is responsible to provide easy access to the health services and information [30,31] such as designing health facilities that help patients find their way, using easily understood signage and symbols in language, facilitating navigational inquiries, co-locate and integrating multiple services in the same facility, help patient to better understand what health care service and benefits are offered, assist people to arrange appointments with other health providers, maintain user-friendly communication skills, develop electronic health applications based on friendly design and patients with limited HL [12,30,31]. According to Egbert and Nana (2009), “the most obvious way to work on limited HL is to have more access to information” [42,43,44]. Although the use of patient portals and other online resources has increased access to health information, patients with limited HL rarely use the internet to obtain health information [45]. This places more responsibility on health care providers to ensure that patients with limited HL have the right health information to make informed health decisions. Patients with limited HL almost always have difficulty understanding written health information [46,47,48] and the inability to read and understand written health information leads to poor health outcomes and reduced quality of patient interactions with service providers [29]. Therefore, all forms and documents should be designed in a simple and understandable language and made available to clients, and by receiving feedback from clients, ensure that the written materials are easily understood by clients [46].

### 4.8. Media Variety

The eighth attribute of the OHL that was obtained in this study is the media variety, which has been introduced as one of the attributes of the OHL in three tools (AHRQ, OHLO, and HLHO-10) [6]. Regarding the media variety sub-theme, two subcategories were obtained under the headings, ‘media design and distribution’ and ‘optimal media features’. These two subcategories contain 12 codes, which are: ‘preparation of educational materials using HL strategies in different formats and their distribution through several channels’, ‘availability of various media for clients with different levels of HL’, ‘existence of educational media in the common languages of the clients’, ‘limited number of messages and use of short sentences’, ‘clear and understandable messages’, ‘use images’, ‘economic cost-effectiveness of the media’, ‘prepare based on HL strategies’, ‘fits the characteristics of the audience’, ‘be reliable’, ‘create attention’, and ‘up to date’. Ryan et al. (2014) evaluated the readability of written health materials distributed to patients with limited HL at an academic science center and found that 29% of the evaluated materials were unsuitable for patients with limited HL [49]. Since the high level of education alone does not reflect the HL status of patients, patients with above-average reading skills and those with a university degree may have difficulty understanding certain medical terms [50].

Therefore, OHLs should provide educational materials using HL strategies in different formats and distribute them to clients through several channels. This helps client to use health information depending on their abilities and preferences [42,43,44].

In 2009, a guide on creating easy-to-understand and easy-to-use health materials was published, which contained important recommendations for providing HL training materials for patients, here are some of these recommendations: start with the most important information to increase understanding, avoid giving too much information because it is easier to understand short messages, use pictures if necessary, try to avoid the use of medical or other technical terms, and consider the format of the document, such as using large fonts, bold headings, standard font styles, and a lot of white space, another important piece of advice is to consider the needs of specific patients to ensure that the material is appropriate for the target audience [12,42,43,44]. In some cases, obtaining feedback from patients or other members of the community will be helpful in designing, implementing, and evaluating educational materials for the patient [51]. This content is consistent with the codes related to the subcategory of desirable media features in our study.

### 4.9. The Role of the Organization in Critical Situations

High-risk situations are the ninth attribute of an OHL [29]. In the qualitative study conducted, the participants’ opinion was that instead of using risk terms, the term critical conditions should be used. Hence, the term critical condition is used in coding. Amongst the three tools (AHRQ, OHLO, and HLHO-10) [6,52], high risk is introduced as one of the attributes of OHLs. This qualitative study was conducted at the time of the COVID-19 epidemic, so participants often generalized critical and high-risk conditions to the COVID-19 epidemic and considered the conditions as risk conditions for health care organizations. For the role of the organization in critical situations, three subcategories were obtained under the headings: ‘provide understandable information’, ‘supplies’ and ‘provide training’, which includes seven codes, which are: ‘continual and transparent information in critical situations’, ‘ensure that informed consent forms are comprehensible to clients’, ‘existence and use of teaching aids’, ‘providing facilities based on the needs of individuals’, ‘educate employees about critical situations’, ‘educate clients in critical situations’ and ‘educate employees about critical situations’.

In critical situations, the organization’s clients should be provided with easy-to-understand information and their HL levels should be improved. Improving HL in critical situations means that health organizations carry out processes to help patients in critical situations [42,43,44]. When dealing with important issues, HL skills may be greatly reduced, because patients are often very emotional, more stressed, and may have difficulty paying attention to their health. In times of crisis, promoting the use of HL practices ensures that patients have clear and understandable information when making decisions in times of distress. Patients dealing with chronic illnesses who need to make important care decisions often need to understand informed consent forms [29]. Therefore, the OHL should design the forms in an easy-to-understand manner and ensure that these forms are easy to understand [16]. Hence, in addition to comprehensible information, appropriate media and training to the crisis and, and understandable forms should be provided to clients in crisis situations. We must also educate employees about emergency situations in order to interact with customers, and provide facilities according to individual needs.

### 4.10. Costs

Cost is the main attribute of the OHL because of its effect on conveying information about health plans and health care bills, (such as co-payments, out-of-pocket expenses), and health insurance [3]. Amongst the four tools (AHRQ, OHLO, HLHO-10, and Org-HLR) cost is also considered as one of the attributes of an OHL [15]. The cost includes two sub-categories ‘information about costs’ and ‘the provision of necessary funds for HL activities in the organization’. Four codes were also derived from cost including ‘clearly inform clients about costs before providing any service’, ‘providing information to clients about the amount of insurance coverage’, ‘allocate specific budgets to support HL activities’, and ‘existence of forms of attracting public aid in order to finance HL’. Therefore, information about costs should be considered in the organization’s agenda, and its rule should be implemented in a clear and understandable way.

In OHL, information about costs should be on the organization’s agenda, and this information should be performed in a clear and understandable way. In the past few years, the healthcare industry has undergone tremendous changes. Numerous health insurance companies have proposed a lot of health insurance plans that contain specific guidelines on underwriting services, joint payments, deductions, and how to file a claim. For patients with adequate HL skills, understanding health insurance plans can be challenging. Therefore, when interpreting health plan information, it is recommended that healthcare providers consider the importance and complexity of the information to improve understanding [53,54]. Therefore, the OHL should have clear information about the amount of expenses covered by insurance for clients. Carrying out the activities of the OHL, of course, has costs, and in the OHL, a specific budget must be considered to support the activities of the HL [12,17]. Of course, this budget allocation will be compensated after a while because the implementation of HL in the organization will reduce costs. Because improving patient outcomes reduces unnecessary use of medical services, the cost of medical care is also reduced [55,56]. In any case, a specific budget must be considered for the HL activities in the organization. In order to provide the necessary funds for the activities of the OHL, a form can be prepared to attract public assistance in the organization and provide it to people who are interested and able to participate in this matter.

### 4.11. Essential Work for the Future

Although the approach of Organizational Health Literacy receives support from Iranian policymakers and healthcare stakeholders, an inadequate level of HL affects large parts of the Iranian population and contributes to unpleasant health outcomes such as limited skills to manage own health and take medication properly, more use of emergency care and hospitalizations [11,12,57]. It seems that shifting health care systems toward effective OHL is a complicated process. According to our findings, 10 main attributes of OHL were defined that characterize an OHL as explained in Table 2. These attributes include the management, integration of HL in the organization, workforce, participation, range of HL skills, HL strategies, access, media variety, the role of the organization in crisis, and costs. The authors argue that there is a need to consider HL as an organizational priority in the healthcare services and a well-designed intervention based on HL strategies should be developed at the organizational level to be responsive. So far, study on OHL reveals that to become health literate, future studies need to include plan organizational self-assessment to better understand HL-based facilitators and barriers, address the relevant attributes of an OHL by emphasizing the integration of HL in the organization structures and workforce, training their health providers in practical communication skills, design visual/audio/written health information tailored to the different levels of HL and patient’s HL needs, and design monitoring systems that identify needs of the patient and the efficacy of interventions programs based HL strategies from the patient perspective.

In this process, different health professionals are vulnerable groups who are considered as change agents because they are at the forefront of educating patients and have the main role in taking over pivotal responsibilities to facilitate organizational change to an OHL [11]. Therefore, there is the need for encouraging health professionals to attend in the implementation of “Health Literacy Universal Precautions”, which aims to promote health professionals’ skills and ability to approach all patients who are have limited information related to improving their own health. Future studies need to examine how different health providers in Iranian healthcare services might contribute to organizational change.

## 5. Limitations

This study was conducted in the time of the COVID-19 pandemic, so due to traffic restrictions and quarantine in Iran, the interviews were conducted by telephone. Likewise, all participants in this study were specialists and staff of health care centers and their job conflicted with the prevalence of COVID-19; therefore, the interviews were conducted slowly. Another limitation was finding terms and concepts appropriate to the Persian language and culture for the interviews because no studies have been conducted on the OHL in Iran.

## 6. Conclusions

The purpose of this study was to determine the attributes of the OHL in health care centers in Iran. In this study, a main theme titled “attributes of Health Literacy Organizations” and 10 sub-themes of attributes of Iranian OHLs were obtained. The ten main attributes of OHL identified in Iran included management, integration of health literacy in the organization, workforce, participation, range of HL skills, HL strategies, access, media variety, and the role of the organization in crisis, and costs. A qualitative research process for the first time attempted to describe the attributes and capabilities of OHL in Iran. Hence, this study was able to draw the attributes of the OHL in accordance with the culture of Iranian society and pave the way for health care organizations in Iran to form an OHL. These attributes may guide the planning of health care centers improvements, and has the potential to promote health service reforms and public health policy.

## Figures and Tables

**Figure 1 ijerph-19-02310-f001:**
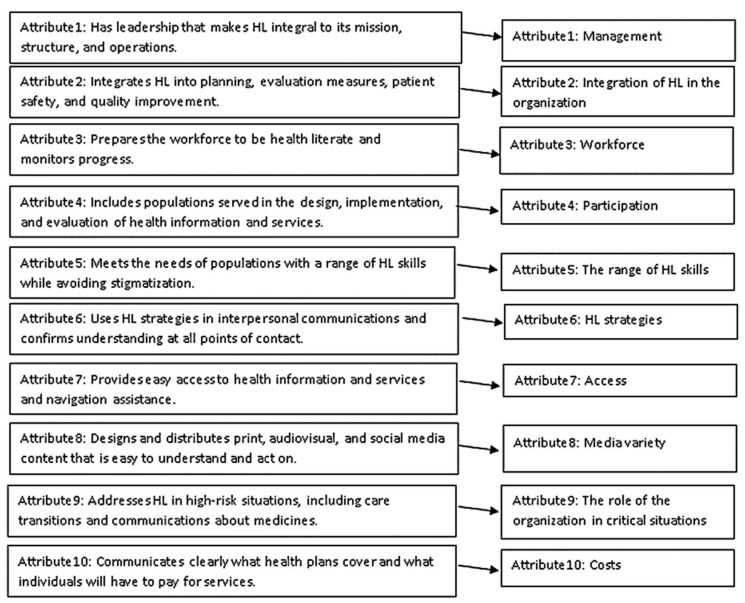
Changes from attributes of the OHL (provided by Brach et al.) to sub-themes (quality content analysis).

**Table 1 ijerph-19-02310-t001:** Characteristics of the study participants.

Participant Characteristics (*n* = 26)		Number (%)
Gender	Male	13 (50)
	Female	13 (50)
Profession	Faculty members	13 (50)
	physicians	2 (7.6)
	Pharmacist	1 (3.8)
	Midwife	2 (7.6)
	Nurse	2 (7.6)
	Health care providers	5 (13.8)
	Security guards	1 (3.8)
Education	PhD	15 (57.6)
	Doctor	1 (3.8)
	Master	5 (19.2)
	Bachelor	4 (15.3)
	Diploma	1 (3.8)
Major	Health education	8 (30.7)
	Health promotion	2 (7.6)
	Community Health	1 (3.8)
	Nursing	2 (7.6)
	Midwifery	2 (7.6)
	Health services management	2 (7.6)
	Pediatrician	1 (3.8)
	Public health	3 (11.5)
	General Practitioner	1 (3.8)
	PhD in Traditional Medicine	1 (3.8)
	Pharmacology	1 (3.8)
	Diploma of Welfare	1 (3.8)
	Accounting	1 (3.8)
Work experience (years)	3–13	9 (34.6)
	14–24	11 (42.3)
	25–35	6 (23)

**Table 2 ijerph-19-02310-t002:** Content analysis of the health professionals and the workforce of health care centers about the attributes of the OHL in Iran.

Sub-Theme	Subcategory	Codes	Meaning Unit
Management	Manager Properties	Appreciate employees who implement HL in the organization	*“The most important thing is that someone is at the top who cares about this work, understands this work, values this work, he should be valued, if the manager does not do this, the staff will not do HL.”* (Participant12, PhD)
		Encourage other organizations to use HL	*“The manager of the organization should collect the results of their work and submit them to other organizations to encourage them to take action to implement HL. By carrying out such activities, we can have a healthy society.”* (Participant 26, Expert)
	Tasks of the manager	Appointment to supervise the implementation of HL	*“It is necessary that in the centers that work in the name of OHL, there must be a person as a supervisor to ensure that HL is implemented in order to supervise the staff for the implementation of HL.”* (Participant 22, MA)
		Handling people’s complaints	*“Complaints of employees and clients to the OHL should be heard by the management of the organization and he should address the problems and complaints so that the OHL can be promoted.”* (Participant 26, Expert)
Integration of HL in the organization	Integrate HL into the structure of the organization	HL in goals and statements	*“First of all, the mission of an OHL must be HL. If you see this, the next steps will be taken.”* (Participant 15, MA)
		Guidelines for promoting employee HL	*“In an OHL, it is necessary to have a specific structure. This structure defines a series of frameworks and standards for them, and can formulate appropriate HL plans for them, and design specific guidelines for improving employees’ HL.”* (Participant 10, PhD)
		Guidelines for promoting people HL	*“The OHL is an organization in which HL is practiced. In an OHL, there are guidelines to promote the HL of people in the organization.”* (Participant 5, PhD)
	Integrate HL into the functioning of the organization	Documents on the implementation of HL	*“The implementation of HL in an OHL should become a culture and be implemented, and the documents for these activities must be prepared and exist in the organization.”* (Participant 21, MA)
		Documents for assessment of HL activities	*“In the OHL, there should be a special program for assessing HL activities, so that there is no specific application, it cannot be assessed how much HL has been upgraded in the community.”* (Participant 26, Expert)
Work force	Education	HL education to employees	*“All the organization’s employees need to train from the lowest level to the highest level, each with their own language, with a simple language.”* (Participant 7, PhD)
		Individuals are responsible for the HL education of employees	*“In view of the fact that education has been forgotten in our health center and even in our health education department, there must be a person responsible for educating employees on OHL and continuing employee training.”* (Participant 21, MA)
		Train communication strategies to employees	*“The workforce before you hire, you need to talk about HL and related communication strategies for these.”* (Participant 3, PhD)
		Staff access to HL content	*“Another issue is the employees’ access to HL training materials, for example, in the hospital wards have a series of summarized trainings using HL strategies that nurse study and teach her patient.”* (Participant 18, Expert)
Workforce	Employment	Employing staff fluent in the clients’ common languages	*“Due to the spoken language of the client, the organization must have persons who can facilitate these exchanges, so it is necessary to hire Multilingual employees.”* (Participant 24, PhD).
		Employing staff familiar with HL	*“The first characteristic of an OHL is that it is the foundation of HL, which means that if I want to employ labor for my clinic, I have to select people whose ability is to promote the HL of the organization and its clients.”* (Participant 5, PhD).
		Existence of HL in the job description of employees	*“First of all, it should be included in the job descriptions of all health literacy staff.”* (Participant 9, PhD).
Participation	People’s participation	Survey of clients regarding the choice of treatment methods	*“Unfortunately, the patient’s participation in the treatment is not monitored, while if the patient’s participation in the choice of treatment is attractive, the treatment responds better.”* (Participant 18, Expert).
		Planning to employ volunteers and health liaisons	*“At the OHL, we must harness the potential of volunteers to assist staff in training.”* (Participant 13, PhD)
	Employee participation	Existence of instructions for employee participation in HL planning	*“When the employees of the organization are involved in planning the activities and their implementation, the work is done better, and if the employees of the organization themselves participate in the evaluation of the works, they can do the previous two steps better, so the employees of the organization should do it. There should be clear guidelines for staff involvement in these cases.”* (Participant 20, Expert)
		Existence of instructions for employee participation in implementing HL	
		Existence of instructions for employee participation in HL assessments	
The range of HL skills	Essential HL skills in relation to clients	Welcoming and evaluating the needs of clients upon entering the organization	*“At the beginning of the client’s entry into the organization, we should welcome the client and assess the needs and assess his level of HL and make sure that his needs are met when leaving? There should be instructions for all this.”* (Participant 25, MA)
		Ensuring that the client’s needs are met when leaving the organization	
	Needs assessment using HL skills	Gather essential information from clients (only once in the organization)	*“Identity, demographic and routine information that is taken from clients to organizations should be taken once that is not boring for clients.”* (Participant 21, MA)
		Identify and guide clients in need of additional assistance	*“Such patients who need extra help also need more follow-up. For example, when we see that this patient is illiterate, we should pay more attention to him than educated people. Hence, these issues should be addressed in the OHL.”* (Participant 25, MA)
		Determining the HL status of clients	*“Determining the health literacy status of clients must be done, because sometimes, what we teach may be too trivial and superficial for the client and he may not listen, and we must first monitor, to see how much his previous information and what are its flaws and misinformation? Then, we will teach people based on it.”* (Participant 22, MA)
HL strategies	Observance of HL strategies by employees in the field of verbal interaction with clients	Use simple, clear and understandable language	*“It is best to talk to the clients in simple language. If you need to use professional terms, please use the translation of these terms and explain to him, so that he can understand.”* (Participant 21, MA)
		Talk to clients with appropriate voice and medium speed	*“The tone of the voice should be neither too loud nor too slow. It should be such that it can hear the other side of the voice well, and it should be gentle and at the same time speak in a counted way so that the other side understands what it is saying.”* (Participant 25, MA)
		Ensure you understand the content provided by clients by getting feedback	*“We have techniques in health education that are good techniques, one of them is feedback, that is, when you explain to a person, ask, see if he understands how much he understands your content? From this simple technique, we can use it in the OHL.”* (Participant 13, PhD)
		Answering clients’ questions and encouraging them to ask questions	*“Clients’ questions should be answered in a good manner in the OHL, and clients should be encouraged to ask questions so that they can raise their knowledge through the question and subsequently increase their HL.”* (Participant 21, MA)
		Allocate enough time for each interaction	*“Setting the right time for each interaction allows the patient to raise and answer their problems and needs and at the same time remain to teach him, so there must be proper communication with the patient and enough time for each interaction in the OHL.”* (Participant 26, Expert)
		Provide training clearly by stating the main and important points	*“Sometimes the trainings are so detailed and things are said that are not useful for the trainee, that is, they are not useful, so in the OHL, it should be noted that the trainings are practical and the main and important points are stated in them.”* (Participant 15, MA)
	Observance of HL strategies by employees in the field of support and writing with clients	Provide training according to the economic ability of clients	*“For example, we do not tell people living in low-income areas to eat protein three times a day, while he may not be able to eat protein once a month. The information we provide is appropriate for their economic situation.”* (Participant 14, MA)
		Provide written training to clients according to their characteristics	*“In written training, attention must be paid to HL issues, and these trainings must be diversified in the organization so that we can provide them according to the characteristics of the audience.”* (Participant 26, Expert)
Access	Access to services and buildings	Easy access to information about the organization and services provided	*“That is, the OHL should be such that people are aware of its location and the type of services it provides, and that the needs of the people are met there.”* (Participant 14, MA)
		Use understandable boards and guides in the organization	*“In designing boards in the OHL, more care must be taken so that people with different characteristics can easily use the boards in navigation.”* (Participant 20, Expert)
		Appropriate number and arrangement of chairs in the organization	“*There should be enough seats for the client to use.”* (Participant 21, MA)
		Easy access to appointment scheduling methods and their accuracy	*“OHLs should make it easy for clients to accept visit appointments, and people in the organization should not be delayed.”* (Participant 24, PhD)
Access		Considering a suitable space for parking vehicles	*“In an OHL, a suitable parking space should be considered so that clients will not have a hard time finding a parking place.”* (Participant 21, MA)
		Possibility of clients’ access to valid educational resources	*“An OHL is an organization that has access to health information resources, credible sources of HL, it is possible to obtain this information.”* (Participant 3, PhD)
		Sending the needy to help and charity centers	*“In OHLs, more help should be provided to people with financial difficulties, and there should be special instructions in this regard.”* (Participant 20, Expert)
	Access to understandable and executable information	Design all forms and documents in simple and understandable language	*“I think the forms in the OHL should be in simple language and understandable, it is much easier for the clients.”* (Participant 16, Diploma)
		Evaluate all content distributed in the organization by getting feedback from clients	*“Before they want to publish their information and media, they need to check to see if their audience understands it or not, that is, to make sure it is understandable.”* (Participant 2, PhD)
		Preparation of forms and documents in the common language of the clients	*“It is better to prepare the forms in the common language of the clients and implement HL strategies in them, and then get feedback from them about whether the contents are understandable for them.”* (Participant 21, MA)
Media variety	Media design and distribution	Preparation of educational materials using HL strategies in different formats and their distribution through several channels	*“In the OHL, there should be educational media in the common languages of those who refer to the centers, and the principle of diversity should be considered in these media, especially if it is in line with HL strategies.”* (Participant 15, MA)
		Availability of various media for clients with different levels of HL	
		Existence of educational media in the common languages of the clients	*“In the OHL, there should be educational media in the common languages of those who refer to the centers, and the principle of diversity should be considered in these media, especially if it is in line with HL strategies.”* (Participant 26, Expert)
	Optimal media features	Limited number of messages and use of short sentences	*“Messages should be short and concise according to the needs of the people, according to the level of information of the people.”* (Participant 25, MA)
		Clear and understandable messages	*“Too simple and schematic, clear and understandable, and too fast for the other person to sit down and read a long text.”* (Participant 19, General Practitioner)
		Use images	*“In designing media, use images that people can understand better, because images tell everything. But the explanation may not be like the image.”* (Participant 4, PhD)
Media variety	Optimal media features	Economic cost-effectiveness of the media	*“Another point that should be considered in the preparation of the media is that the media produced in the organization should be as economical and cost-effective as possible, but it should be borne in mind that this cost-effectiveness should not reduce the capabilities of the media or it should question the other characteristics of the desired media, but in addition to observing other points, it should also be considered economically viable.”* (Participant 26, Expert)
		Prepare based on HL strategies	*“Another feature is the preparation of this medium based on HL strategies, which, especially in terms of writing, the items that are considered HL have been observed in it.”* (Participant 9, PhD)
		Fits the characteristics of the audience	*“In media design, understand this, take into account what psychological characteristics, characteristics and differences the target group may have in terms of demographics and social needs, and the media should be appropriate to the characteristics of the audience.”* (Participant 10, PhD)
		Be reliable	*“An OHL is an organization that can access health information resources, effective resources, and creditable resources. It means that there are credulous media in the OHL.”* (Participant 3, PhD)
		Create attention	*“You watch TV commercials, about a junk food item like puffs, they usually use the best commercials, then for HL training, nothing attractive is used that one wants to follow the training.”* (Participant 1, PhD).
		Up to date	*“The media in the HL organization must be up to date, we do not use modern methods at all.”* (Participant 1, PhD).
The role of the organization in critical situations	Provide understandable information	Continuous and transparent information in critical situations	*“The mission of an OHL in dangerous situations is to provide continuous and transparent information about dangerous situations so that it can take steps to win the trust of the audience.”* (Participant 15, MA)
		Ensure that informed consent forms are comprehensible to clients	*“In the OHL, it must be ensured that the forms, especially the informed consent forms, are such that a person with low information and low HL can understand.”* (Participant 23, Expert)
	Supplies	Existence and use of teaching aids	*“We should use teaching aids in our training, especially in cases where there are special conditions, for example, a box of pills can be used to teach when to take pills, and it is very helpful.”* (Participant 21, MA)
		Providing facilities based on the needs of individuals	*“We must provide services to those who are most in need, away from the considerations of kinship, ethnicity, ethnicity, religion, and nationality....”* (Participant 6, PhD)
	Provide training	Educate employees about critical situations	*“In critical situations, there should be a review in each position, i.e., the health education unit of the health center shall determine retraining courses for its health education experts.”* (Participant 10, PhD)
		Educate clients in critical situations	*“For example, in critical situations such as corona, we need to teach customers how to protect themselves from viruses.”* (Participant 14, MA)
		Preparation and distribution of media appropriate to critical situations	*“OHLs should use appropriate media in critical situations, for example, in the event of a corona epidemic, a video of a corona patient should be shown. This is much more effective than someone talking about the disease for two hours…”* (Participant 4, PhD)
Costs	Information about costs	Clearly inform clients about costs before providing any service	*“As people become more aware of costs and use different resources to meet their health needs, people will find better management and decision-making methods, and ultimately become more satisfied with the service.”* (Participant 15, MA)
		Providing information to clients about the amount of insurance coverage	*“The amount of insurance costs should be clear to the individual, i.e., depending on the insurance that the person is covered, this information should be given to him how much he insures and how much he has to pay out of pocket.”* (Participant 21, MA)
	The provision of necessary funds for HL activities in the organization	Allocate specific budgets to support HL activities	*“Allocating a specific budget to support HL activities in the organization is one of the ways to provide the necessary funding for HL activities.”* (Participant 26, Expert)
		Existence of forms of attracting public aid in order to finance HL	*“In the areas where we worked, before in the south of Zahedan province, the housing of the people was unsuitable, but the mosques were very, very stylish, because there was public aid for the mosques. As for the OHL, if the people’s help is attracted in a proper way, the necessary funds will be provided for HL activities.”* (Participant 6, PhD).

## Data Availability

No new data were created or analyzed in this study. Data sharing is not applicable to this article.

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
