# Peer review of "Attributes of Organizational Health Literacy in Health Care Centers in Iran: A Qualitative Content Analysis Study"

_ijerph, 2022, doi:10.3390/ijerph19042310_

Round 1

Reviewer 1 Report

The authors investigated Organizational Health Literacy (OHL) in Iran, where less information is available regarding the healthcare system. Using recent semi-structured interviews, they identified ten major attributes of OHL, including management, integration of health literacy in the organization, workforce, participation, skills, strategies, access, media variety, the role of the organization in crisis and costs.

The article is interesting, but the authors should clarify the followings:

Introduction

The subheading   “Theoretical Framework” should be removed.

Method

-In study setting, it has been mentioned that this study was conducted in Mashhad. However, in other parts of methods, it has been mentioned that the researchers gathered information from different cities of Iran. This should also be clarified in the study setting.

- 2.5. Research questions

-“He then used semi-structured questions during the interview”

Just one interviewer (He)? Should it be they instead of he?

Result

Please clarify these sentences

-The study involved 26 people, half of whom were female interviewers and

- In this study, (57.6) participants had a  doctorate degree and (30.7) participants had a health education.

Please remove parentheses

-Table 1,

Mama,  Please write it in English not Persian

-Please remove parentheses

The average duration of the interview was (44.8±13.04)

units are shown in (Table 2, Figure 1).

Discussion

Discussion section should be shorter.

Author Response

Reviewer 1

Dear Dr/ Prof,

We would like to thank the reviewer for careful and thorough reading of this manuscript and for the thoughtful comments and constructive suggestions, which help to improve the quality of this manuscript. We tried to answer to all valuable comments/suggestions/queries and all answers to comments are highlighted within the document by using Yellow color.

The authors investigated Organizational Health Literacy (OHL) in Iran, where less information is available regarding the healthcare system. Using recent semi-structured interviews, they identified ten major attributes of OHL, including management, integration of health literacy in the organization, workforce, participation, skills, strategies, access, media variety, the role of the organization in crisis and costs. The article is interesting, but the authors should clarify the followings:

Introduction

Comment 1: The subheading  “Theoretical Framework” should be removed.

Reply 1: it was removed

Method

Comment 2: In the study setting, it has been mentioned that this study was conducted in Mashhad. However, in other parts of the methods, it has been mentioned that the researchers gathered information from different cities of Iran. This should also be clarified in the study setting.

Reply 2: it was corrected. This study was conducted on 26 participants from different cities (Mashhad, Tehran, Tabriz, Yasuj, Maragheh, Bojnourd, Zahedan, Torbat Heydariyeh, Kurdistan, and Chenaran) from June 2020 to January 2021

Comment 3: Research questions: -“He then used semi-structured questions during the interview”

Just one interviewer (He)? Should it be they instead of he?

Reply 3: This sentence was corrected as follows:  The researcher's first question from the participants was, what do you think about the OHL-related attributes? Then, semi-structured questions were used during the interview.

Result

Comment 4: Please clarify these sentences

-The study involved 26 people, half of whom were female interviewers and

- In this study, (57.6) participants had a doctorate degree, and (30.7) participants had a health education.

Reply 4:  it was corrected as follows: The study involved 26 people, half of whom were female interviewees and other participants were members of health faculty (Table 1). In this study, 57.6% of participants had a doctorate degree and 88.6% of participants had a health education

Comment 5: Please remove parentheses: Done

Comment 6: Table 1,

  • Mama, please write it in English not Persian: Done
  • -Please remove parentheses: The average duration of the interview was (44.8±13.04): Done

Discussion

The discussion section should be shorter: Done

Reviewer 2 Report

My comments are as follows: 

  1. In the introduction, author forgot to mention WHY is it important for Iranian to improve their HL. Such as making better healthcare decision for their own health, which in turn, have better response to treatment regime.
  2. In section 1.2, the repetition of 'these 10 attributes' in these few sentences, is rather awkward: "The theoretical framework of this study is the 10 attributes of OHL expressed by 58 Brach et al (3). these ten attributes can be seen in several tools of the OHL. These ten at-59 tributes are the basis for conducting a qualitative study of the present directed content 60 analysis. These ten attributes are:...." please rephase.
  3. Also in section 1.2, don't use the same bracket for 'these 10 attributes' and the in-text numeric citation as it might cause confusion.
  4. the questions in '2.5. Research questions' can be expressed in bullet form instead for better presentation.
  5. Please include '%' for (57.6) and (30.7) in line 127-128.
  6. Is it 'interviewers' or 'interviewees' in line 126 "female interviewers"? If it is 'interviewer', it is fair for them to participate in the study?
  7. I don't think is correct to say only 30.7% of their participants have health education when 10/12 major is healthcare related. Hence, this is a false assumption.
  8. Likewise for line 342-345, avoid using the same bracket for intext numeric citation and US centers for disease control and prevention's recommendation.
  9. Please include a section for the possible future work of this study.

Author Response

Reviewer 2

Dear Dr/ Prof,

We would like to thank the reviewer for careful and thorough reading of this manuscript and for the thoughtful comments and constructive suggestions, which help to improve the quality of this manuscript. We tried to answer to all valuable comments/suggestions/queries and all answers to comments are highlighted within the document by using Red color.

Comment 1: In the introduction, author forgot to mention WHY is it important for Iranian to improve their HL. Such as making better healthcare decision for their own health, which in turn, have better response to treatment regime.

Reply 1: based on this comment following information was added as follow:  their finding highlighted some of the major challenges and gaps in the field of HL in Iran such as the lack of a clear and fixed definition for HL, lack of OHL-related attributes and guidelines in health services to manage patient based on their HL levels, no locally comprehensive screening instrument for HI in Iran, and poor patient-health provider communication skills. In Iran, promoting HL is necessary to provides the base on which Iranian population are enabled to improve their own health through making better decision related to their health condition, involve successfully with self-care skills and health community action, and push authorities to meet their duties in addressing health equity. Meeting the HL needs of Iranian population will promotes progress in decreasing health inequity. Efforts to improve HL is crucial in whether the environmental, economic, and social promotion are fully realized. It seems that in order to improve the HL level of the Iranian people, HL should be considered as an organizational priority in the health service and interventions should be carried out at the organizational level. Results indicated HL is not still integrated into the health organization’s strategic planning and vision. Therefore, health policy makers and researchers face more challenges in what they recognize about attributes of the OHL and what steps they should use OHL-related attributes and guidelines to improve HL skills. Since no action has been taking in Iran for the OHL, this study aims to explain the attributes and content of the OHL in health care centers in Iran. Such information help health services to effectively implement OHL-related attributes that improve the quality of health service and personal health literacy skills in community

Comment 2: In section 1.2, the repetition of 'these 10 attributes' in these few sentences, is rather awkward: "The theoretical framework of this study is the 10 attributes of OHL expressed by 58 Brach et al (3). these ten attributes can be seen in several tools of the OHL. These ten at-59 tributes are the basis for conducting a qualitative study of the present directed content 60 analysis. These ten attributes are:" please rephase.

Reply 2: it was corrected

Comment 3: Also in section 1.2, don't use the same bracket for 'these 10 attributes' and the in-text numeric citation as it might cause confusion.

Reply 3: it was corrected based on this comment in all part of manuscript.

Comment 4: the questions in '2.5. Research questions' can be expressed in bullet form instead for better presentation.

Reply: Done

Comment 5: Please include '%' for (57.6) and (30.7) in line 127-128.

Reply 5: Done

Comment 6: Is it 'interviewers' or 'interviewees' in line 126 "female interviewers"? If it is 'interviewer', it is fair for them to participate in the study?

Reply 6: it was interviewees, however this sentence was corrected based on following format: The researcher's first question from the participants was, what do you think about the OHL-related attributes? Then, semi-structured questions were used during the interview.

Comment 7: I don't think is correct to say only 30.7% of their participants have health education when 10/12 major is healthcare related. Hence, this is a false assumption.

Reply 7: it was corrected

Comment 8: Likewise, for line 342-345, avoid using the same bracket for in text numeric citation and US centers for disease control and prevention's recommendation.

Comment 8: Please include a section for the possible future work of this study.

Reply 8: it was explained as follow:

Although the approach of organizational health literacy receives support from Iranian policymakers and healthcare stakeholders, an inadequate level of HL affects large parts of the Iranian population and contributes to unpleasant health outcomes such as limited skills to manage own health and take medication properly, more use of emergency care and hospitalizations. It seems that shifting health care systems toward effective OHL is a complicated process. According to our findings, ten main attributes of OHL were defined that characterize an OHL as explained in Table 2. These attributes include the management, integration of HL in the organization, workforce, participation, range of HL skills, HL strategies, access, media variety, the role of the organization in crisis, and costs.  The authors argue that there is a need to consider HL as an organizational priority in the healthcare services and a well-designed intervention based on HL strategies should be developed at the organizational level that is, be responsive. So far, study on OHL reveals that to become health literate, future studies need to include plan organizational self-assessment to better understand HL-based facilitators and barriers, address the relevant attributes of an OHL by emphasizing the integration of HL in the organization structures and workforce, training their health providers in practical communication skills, design visual/audio/written health information tailored to the different levels of HL and patient's HL needs, and design monitoring systems that identify needs of the patient and the efficacy of interventions programs based HL strategies from the patient perspective. In this process, different health professionals are vulnerable groups who are considered as change agents because they are at the forefront of educating patients and have the main role in taking over pivotal responsibilities to facilitate organizational change to an OHL. Therefore, there is the need for encouraging health professionals to attend in the implementation of “Health Literacy Universal Precautions”, which aims to promote health professionals’ skills and ability to approach all patients who are have limited information related to improving their own health. Future studies need to examine how different health providers in Iranian healthcare services might contribute well to organizational change.

Round 2

Reviewer 2 Report

The authors have addressed my comments.